# A Cost-Effective Portable Multiband Spectrophotometer for Precision Agriculture

Francisco Javier Fernández-Alonso [1,2], Zulimar Hernández [3] and Vicente Torres-Costa [1,2,3,*]

1 Applied Physics Department, Autonomous University of Madrid, 28049 Madrid, Spain; franciscoj.fernandez05@estudiante.uam.es
2 Nicolás Cabrera University Institute of Materials Science, Autonomous University of Madrid, 28049 Madrid, Spain
3 Copernicus-UAM Remote Sensing Laboratory, Autonomous University of Madrid, 28049 Madrid, Spain; zulimar.hernandez@uam.es
* Correspondence: vicente.torres@uam.es

**Abstract:** The United Nations marks responsible consumption and production as one of the 17 key goals to fulfill the 2030 Agenda for Sustainable Development. In this context, affordable precision instruments can play a significant role in the optimization of crops in developing countries where precision agriculture tools are barely available. In this work, a simple to use, cost-effective instrument for spectral analysis of plants and fruits based on open-source hardware and software has been developed. The instrument is a 7-band spectrophotometer equipped with a microprocessor that allows one to acquire the reflectance spectrum of samples and compute up to 9 vegetation indices. The accuracy in reflectance measurements is between 0.4% and 1.4% full scale, just above that of high-end spectrophotometers, while the precision at determining the normalized difference vegetation index (NDVI) is 0.61%, between 3 and 6 times better than more expensive commercial instruments. Some use cases of this instrument have been tested, and the prototype has proven to be able to precisely monitor minute spectral changes of different plants and fruits under different conditions, most of them before they were perceptible to the bare eye. This kind of information is essential in the decision-making process regarding harvesting, watering, or pest control, allowing precise control of crops. Given the low cost (less than USD 100) and open-source architecture of this instrument, it is an affordable tool to bring precision agriculture techniques to small farmers in developing countries.

**Keywords:** precision agriculture; spectrophotometer; vegetation indices; NDVI; cost-effective; sustainable development

## 1. Introduction

The development and regulation of autotrophic living beings on our planet are controlled by sunlight [1]. The most important sunlight-induced process is photosynthesis, which occurs due to some pigments, mainly chlorophyll, absorbing solar radiation within the visible electromagnetic spectrum range. This gives rise to a set of chained chemical reactions in which vital organic compounds, such as sugars, fats, and proteins are synthesized from simple inorganic compounds such as water, carbon dioxide, and minerals, among others [2,3]. These organic molecules are the building blocks of all autotrophs [4] and, therefore, being able to indirectly monitor how photosynthesis is taking place in a plant allows us to know about its physiological state.

The efficiency in the process of photosynthesis can be affected by various factors, both abiotic, such as drought or excess of water [5,6], temperature [7], salinity [8,9], nutrient deficiencies [10], or agrochemicals [11], and biotic, such as pathogens [12] or insect pests [13]. All of these effects may have implications for vegetation functioning [14,15], and also for the productivity of agricultural holdings [16,17].

The most common pigment type that contributes to photosynthesis is *a*-Chlorophyll, which is found in all autotrophs [18]. Other types of pigments, such as carotenoids, can be found in some plants. They are called accessory pigments since they can only absorb sunlight but are not able to convert it into chemical energy [19,20].

*a*-Chlorophyll exhibits strong absorption in blue and red wavelengths, and moderate and high reflectance in green and near-infrared, respectively [21]. Therefore, by measuring the reflectance spectrum of the leaves, the superficial concentration of *a*-Chlorophyll can be estimated [22,23]. Furthermore, as different types of pigments present slightly different reflectance spectra [21], it is possible to estimate what type of chlorophyll is present in the plant [24,25].

Furthermore, the slight difference in reflectance of fruits and vegetables that can be observed during ripening, such as in tomatoes [26], pears [27], strawberries [28], or peppers [29], can be used, for example, to determine the optimal moment when it is appropriate to harvest a vegetable. Therefore, this could be determined with greater precision than with the bare eye by using a spectrophotometer with sufficient precision.

The result of spectral analysis is commonly summarized in the form of vegetation indices, which relate the reflectance in two or more wavelength intervals, or bands. Some of the most common vegetation indices are shown in Table 1 [3,30]. The most common vegetation index is the normalized difference vegetation index (NDVI), which is often used as an indicator of the chlorophyll content of the sample [31,32]. However, there are many other vegetation indices developed for different applications [32–34].

**Table 1.** Some of the most common vegetation indices.

| Vegetation Index | Equation | Reference |
|---|---|---|
| Simple Ratio (SR) | $SR = \frac{R_{NIR}}{R_{RED}}$ | [35] |
| Normalized Pigment Chlorophyll Ratio Index (NPCI) | $NPCI = \frac{R_{RED} - R_{BLUE}}{R_{RED} + R_{BLUE}}$ | [36] |
| Renormalized Difference Vegetation Index (RDVI) | $RDVI = \frac{R_{NIR} - R_{RED}}{\sqrt{R_{NIR} + R_{RED}}}$ | [37] |
| Normalized Difference Vegetation Index (NDVI) | $NDVI = \frac{R_{NIR} - R_{RED}}{R_{NIR} + R_{RED}}$ | [38] |
| Green NDVI (NDVI$_g$) | $NDVI_g = \frac{R_{NIR} - R_{GREEN}}{R_{NIR} + R_{GREEN}}$ | [39] |
| Blue NDVI (NDVI$_r$) | $NDVI_b = \frac{R_{NIR} - R_{BLUE}}{R_{NIR} + R_{BLUE}}$ | [40] |
| Infrared Percentage Vegetation Index (IPVI) | $IPVI = \frac{R_{NIR}}{R_{NIR} + R_{RED}}$ | [41] |
| Structure Insensitive Pigment Index (SIPI) | $SIPI = \frac{R_{NIR} - R_{BLUE}}{R_{NIR} + R_{RED}}$ | [42] |
| Enhanced Vegetation Index (EVI) | $EVI = \frac{2.5(R_{NIR} - R_{RED})}{R_{NIR} + 6R_{RED} - 7.5R_{BLUE} + 1}$ | [43] |

The continuous measurement of such indices is an especially valuable tool for the study and monitoring of crops, as it provides vital information regarding the need for watering, fertilization, or pest control, among others. Nowadays, there are numerous commercial instruments that can determine some of the above indices. This can be done using multispectral sensors, in which reflectance is measured in a few bands, and the vegetation indices are calculated from the relationships between them, or hyperspectral (spectroscopic) sensors, in which reflectance is continuously determined in a broad wavelength range, and the vegetation indices are then obtained by comparing reflectance at two or more wavelengths [3]. However, the devices nowadays available on the market have some drawbacks: (1) they are relatively expensive (more than USD 2000); (2) most measure only one or two vegetation indices; and (3) the geometry of many of them does not allow to perform measurements on non-leaf objects (such as vegetables) [44–47].

In this work, a cost-effective instrument, based on the Arduino open-source platform [48,49] was designed and implemented. The designed device is capable of measuring reflectance in seven bands between blue and infrared and calculating all the indices shown in Table 1. This low-cost portable instrument is aimed at farmers to provide them with real-time information that can aid them in decision making. In addition, the device was

designed in a geometry that allows samples other than leaves, such as fruits or vegetables, to be measured. This article describes the design of the device, and its performance is compared to that of equivalent commercial instruments. In addition, to test the performance of our device in some cases of practical application, two experimental tests were carried out. On the one hand, changes in the physiological state of plants both when the level of sunlight exposure and soil water are modified were monitored with our device and with a commercial instrument. On the other hand, changes in the reflectance spectra of several fruits during ripening were also monitored.

In this way, we seek to make this technology accessible to farmers in developing countries, favoring compliance with Sustainable Development Goals 2 (Zero Hunger) and 12 (Responsible Consumption and Production) [50].

## 2. Materials and Methods

### 2.1. Optical Design

The device consists of several light-emitting diodes (LEDs) of different wavelengths that are lit sequentially onto the sample. The light reflected from the sample is picked up by a photodiode, amplified, and fed into a microprocessor. The amount of light reflected by the sample at each band is compared to that reflected by a perfectly white reference and a dark reference so that the actual sample's reflectance can be derived. Seven LEDs that cover different bands across the VIS-NIR range were arranged in a circular geometry around a photodiode. A cylindrical shell enclosing the LEDs and the photodiode was 3D-printed, isolating the system from external light sources. The photodiode was, in turn, placed inside a second cylinder to prevent direct illumination from the LEDs. A diagram and a photograph of the device are shown in Figure 1. Several lids with different apertures were also fabricated so samples of different geometries can be measured. The cylindrical shells and the lids were 3D-printed using black polylactic acid (PLA) filament to reduce internal reflection. All the pieces were designed with a rough surface finish to further minimize stray light.

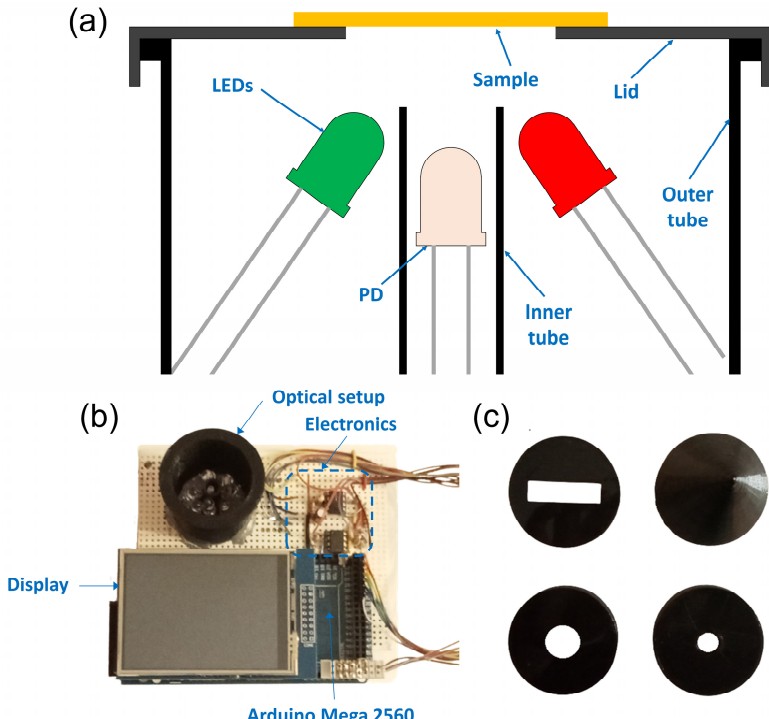

**Figure 1.** (**a**) Optical setup schematic. (**b**) Photograph of the implemented device. (**c**) Photograph of three lids and the cone lid used for the dark reference measurement.

The emission spectrum of each LED was measured using a fiber optic CCD *BLUE-Wave Miniature spectrometer* (StellarNet Inc., Tampa, FL, USA). The photodiode used was a PN junction silicon photodiode model TEFD4300 (Vishay Intertechnology, Inc., Malvern, PA, USA). Figure 2 shows the emission spectrum of each LED, as well as the relative spectral sensitivity of the photodiode, while Table 2 shows the denomination given to each band, its median wavelength $\lambda_{med}$, the wavelength of maximum intensity $\lambda_{max}$, and the half-width at half maximum of the spectrum $\Delta\lambda$. If the use of a different wavelength is required for a specific application, it would suffice to place an additional LED with the appropriate wavelength.

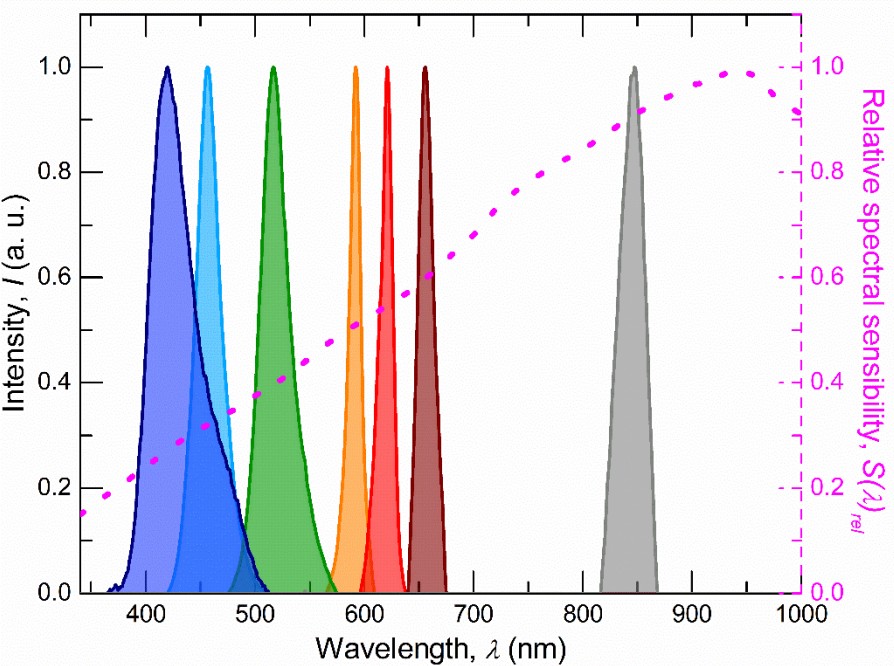

**Figure 2.** Emission spectrum of each of the seven LEDs used. The relative spectral sensitivity of the detector over the wavelength range used is shown by magenta a dashed line.

**Table 2.** Name and characteristics of each band considered.

| Band | $\lambda_{med}$ (nm) | $\lambda_{max}$ (nm) | $\Delta\lambda$ (nm) |
|---|---|---|---|
| Blue-I | 428.5 | 419.5 | 23.5 |
| Blue-II | 457.7 | 457.0 | 13.0 |
| Green | 519.5 | 517.0 | 15.0 |
| Orange | 591.0 | 592.5 | 5.5 |
| Red-I | 620.2 | 621.0 | 7.0 |
| Red-II | 656.2 | 656.0 | 8.0 |
| Infrared | 848.1 | 847.5 | 13.5 |

The fact that not all the LEDs emit the same intensity, and that the detector does not present a constant spectral sensitivity makes it necessary to calibrate the response of the photodetector to each LED in order to derive an actual reflectance value. For that, a white reference that provides a known 100% reflectance in all bands is needed. To this effect, a sheet of polytetrafluoroethylene (PTFE) was used as a white reference. This material is known to have a reflectance close to 100% in the wavelength range of interest, with a deviation of less than 0.3% full scale, as verified with a *JASCO V-560* double beam spectrophotometer (JASCO International Co. Ltd., Hachioji, Japan). On the other hand, a dark reference is required to determine the noise floor of the system. For this, a conical lid made of black polylactic acid (PLA) was used. This allows deducting the dark photosignal due to stray light. The reflectance spectrum of the black polylactic acid (PLA)

was measured to be lower than 1.2% with a deviation in the order of 0.5% in the wavelength range considered.

## 2.2. Signal Acquisition

The acquisition circuit consists of four stages: a first stage corresponding to the LED control, a second stage in which the current generated by the photodiode is converted to voltage, a third low-pass filtering stage, and a fourth adaptive amplification stage. Finally, the amplified signal is fed into a 10-bit analog-to-digital converter (ADC) for numerical processing. A schematic of the signal acquisition circuit is shown in Figure 3.

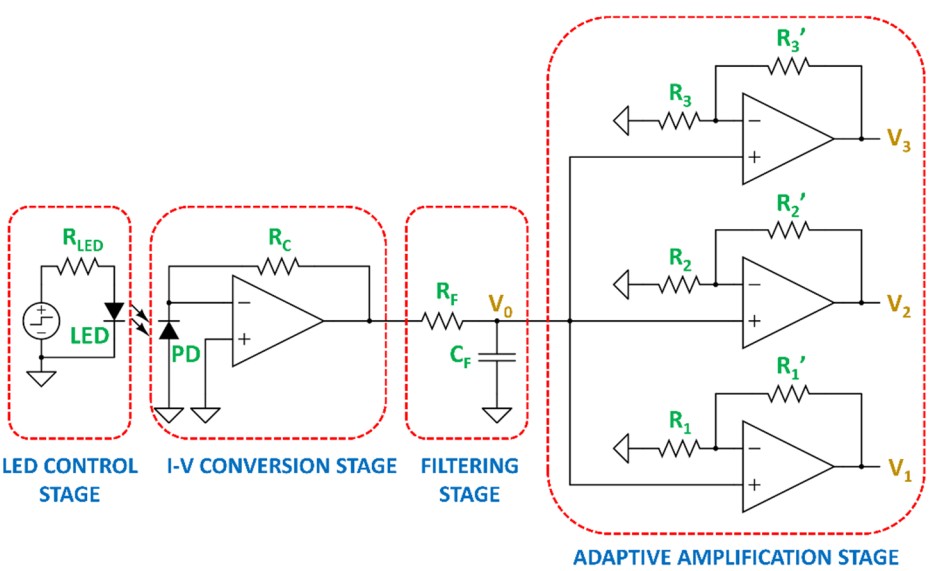

**Figure 3.** Scheme of the signal acquisition circuit.

The first stage consists of each one of the LEDs and its bias resistor. Each LED is directly powered by a 5 V digital signal from the microcontroller. A bias resistor is used to limit the current across the LEDs to 40 mA, which is the maximum current the microcontroller can provide [51].

In the second stage, the photocurrent generated by the photodiode is converted into voltage through a current-to-voltage converter. A low conversion ratio of 82 mV/μA was chosen to avoid a strong impedance mismatch with the following stages. Consequently, the signal at the converter's output is small, in the order of 10–100 mV. Therefore, since the resolution of the analog-to-digital conversion is going to be approximately 5 mV, a further amplification stage is required.

Before amplification, the signal is filtered in the third stage by a passive low-pass filter (LPF). It was determined that the most prominent electronic noise in the signal had a frequency above 1 kHz. Considering that the LEDs are switched on and off at a frequency of around 65 Hz, a passive LPF with a cutoff frequency of 1 kHz was implemented using an $R_F = 160\ \Omega$ resistor and a $C_F = 1.0\ \mu F$ capacitor. In this way, the amplitude of the high-frequency electronic noise is reduced by approximately a factor of 50.

Finally, an adaptive signal amplification stage was implemented. To do this, three non-inverting amplifier circuits of increasing gain were arranged, as shown in Figure 3. The gains of the amplifiers were chosen by measuring a large variety of samples of different color, brightness, and texture, to provide an overview of the possible signal levels to be expected. After these tests, theoretical gains of $G_1 = 13$, $G_2 = 100$, and $G_3 = 300$ were determined to be optimal. Once the amplifiers had been implemented, the actual gain of each one was experimentally determined. At each measurement, the amplifier that provides the highest voltage below saturation is used. In this way, the signal with the highest possible amplitude is fed into the analog-to-digital converter (ADC) every time,

depending on the overall emission intensity of the LED being in use and the reflectivity of the sample being measured.

The whole system is controlled using an open-source Arduino Mega 2560 microcontroller board. This board is provided with digital outputs that allow the LEDs to be switched on and off, as well as with several 10-bit analog-to-digital converters, that allow recording the output voltage of each of the three non-inverting amplifiers simultaneously. Since the input dynamic range of the ADCs is 5 V, the resolution of the analog-to-digital conversion is slightly lower than 5 mV, or about 0.1% full scale. The whole device is powered using a 5 V battery. To implement the current-to-voltage converter, as well as the non-inverting amplifiers, rail-to-rail operational amplifiers model LMC6482 powered at 5 V were used. This type of operational amplifier provides the widest possible output voltage range [52], which allows optimal utilization of the ADCs. In this way, the overall device's resolution is maximized.

The bill of materials for this device, including the Arduino board and a touchscreen color display, is less than USD 65, which is between 10 and 100 times less than the price of devices available in the market for similar applications.

### 2.3. Data Processing

To measure the intensity of the reflected light in a band, the proper gain amplifier must be first chosen, so that the signal is maximal without saturation. To do this, the corresponding LED is turned on and the output voltage of the highest gain amplifier $G_3$ is measured three times by the microcontroller's ADC. Then, the three values are averaged. If the average value is lower than 90% of the ADC's full scale, the amplifier is not saturated and $G_3$ is taken as the optimal gain for that band. If it is higher than 90%, the amplifier is most probably saturated, so the process is repeated with the intermediate gain amplifier $G_2$. If the intermediate amplifier is not saturated, $G_2$ is then taken as the optimal gain. Otherwise, the amplifier with the lowest gain $G_1$ is taken as the optimal for that band.

Once the appropriate gain for each band is chosen, the actual measurement is performed. To reduce the influence of electronic and light noise, a digital synchronous filter (DSF) [53] was implemented in the microprocessor. In a DSF, several successive measurements are taken with the LED on and off. Provided the different noise components have characteristic periods different from the sampling rate of the DSF, the signal-to-noise ratio can be greatly enhanced by subtracting the average of the measurements performed with the LED on and off, respectively. Given that the lower the photocurrent, the stronger the influence of the electronic and light noise, the optimum number of sampling points used in the DSF was optimized for each amplifier circuit. It was considered adequate to implement the DSF with 10-, 20-, and 40-point pairs, when acquiring the signal from the amplifier with the lowest, intermediate, and highest gain, respectively. In any case, the LEDs were turned on and off at 65 Hz. Finally, once filtered, the signal is normalized to the gain of the amplifier used in each case so that all intensities can be directly compared.

Two buttons are constantly displayed on the screen in order to perform calibration measurements with the white or black reference. In the event that a white or black calibration measurement is required, the above procedure is repeated, with the exception that the DSF is implemented with three times more points. This process takes less than ten seconds.

Next, for each band $i$, a coefficient $r_i$, directly proportional to the reflectance of the sample at that band, is calculated. This coefficient is obtained from the sample's signal intensity $I_i^s$, and takes into account the intensity of the reflected light from the dark $I_i^d$ and white $I_i^w$ references for that band. This coefficient is given by Equation (1).

$$r_i = \frac{I_i^s - I_i^d}{I_i^w - I_i^d} \tag{1}$$

Although it is not possible with this setup to determine the absolute reflectance of a sample at each band, since that would require integrating the reflected light in all directions,

it is still possible to make an adequate representation of the reflectance spectrum using the $r_i$ coefficients (Figure 4). This limitation is present in any instrument that is not provided with an integrating sphere [54], but is not relevant for calculating the vegetation indices, since these are based on relative reflectance differences between bands, and not on their absolute values.

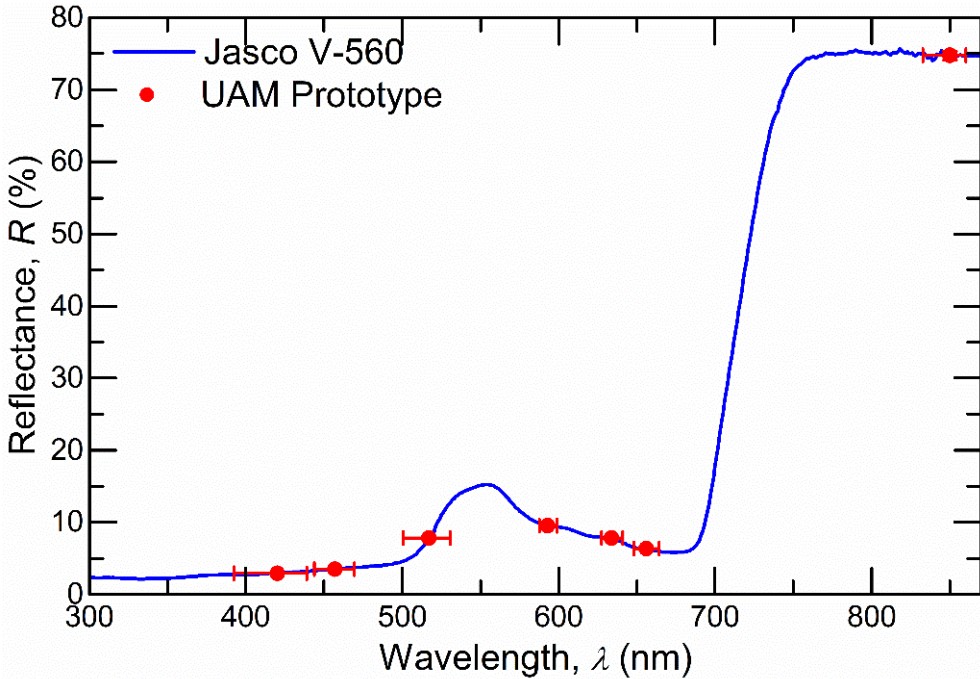

**Figure 4.** Reflectance spectrum of a leaf obtained using a *JASCO V-560* spectrophotometer, together with the reflectance obtained using our prototype for the bands considered.

The time it takes for the device to calculate and update the reflectance spectrum of a sample, as well as the different vegetation indices, depends on the gain used for each band, but in any case, it is between one and four seconds.

## 3. Device Characterization

### 3.1. Accuracy Determination

The accuracy of the device presented, that is, the deviation between the results produced by the device and reality, was studied. For that, a double beam spectrophotometer provided with an integrating sphere, model *JASCO V-560*, with a photometric accuracy of 0.3% on the full scale, was taken as a reference, against which our results were compared.

It is important to note that the reference spectrophotometer measures the reflectance throughout the spectral range considered, with a spectral bandwidth of 2 nm, while our device measures the integrated reflectance in a certain band corresponding to the emission spectrum of each LED. In addition, the sensitivity of the detector varies slightly along each band. Therefore, to properly compare the results of both devices, the reference reflectance value should be taken as the weighted average of the reflectance provided by the reference device at each wavelength $R_\lambda$, weighted by the emission intensity of the LEDs $I_\lambda$ and the relative sensitivity of the photodetector $S_\lambda^{rel}$ at that wavelength. Thereby, the reflectance at each band measured by the reference spectrophotometer $R_i$ is taken as Equation (2).

$$R_i = \frac{\sum_\lambda R_\lambda I_\lambda^i S_\lambda^{rel}}{\sum_\lambda I_\lambda^i S_\lambda^{rel}} \quad (2)$$

The reflectance coefficient $r_i$ from our device and the reference reflectance $R_i$ will be different due to the different optical designs, but they will be proportional in the whole

spectrum. However, this proportionality will be different for each sample depending on how specular or diffuse the sample surface is.

To determine the accuracy, or better, the inaccuracy of our prototype, the reflectance spectra of 75 samples of different color, texture, size and brightness were measured with both the reference spectrophotometer and our prototype. The samples used for this purpose were leaves of various plants and vegetables in different physiological states. Half of the measurements were carried out without a lid, and the rest, using the different lids shown in Figure 1a. For each sample, the proportionality between $R_i$ and $r_i$ was determined by a least-squares fit of all the bands, and the deviation of each $r_i$ from the fit was determined. The results from the 75 analyzed objects were summarized in a histogram of the deviation for each band. By performing a Gaussian fit to each histogram, the standard deviation $\sigma_i$ was taken as the inaccuracy of the device at band $i$.

The mean discrepancy between the reflectance coefficient $r_i$, once corrected by the proportionality fit, and the reference reflectance $R_i$ for each band, as well as the standard deviation, are summarized in Figure 5, normalized to the actual reflectance $R_i$. In addition, Table 3 shows the inaccuracy determined for each band. The accuracy obtained is better than 1.5% of $R$ for all the bands considered, while the average accuracy of the device is 0.78% of $R$. It is observed how the accuracy of the orange band is worse than that of the rest of the bands, showing an inaccuracy about 60% higher than the mean. This can be attributed to the fact that the orange LED used presents a wider emission angle, which may lead to additional errors derived from the reflection of light from the inner sides of the cylindrical shell or the lid used. On the other hand, the accuracy of the infrared band is better than 0.42% of $R$, due to the narrow emission cone of the infrared LED used, and the high sensitivity of the photodiode to that wavelength range. By virtue of the above, it is appropriate to use LEDs with the narrowest possible emission cone when implementing the device.

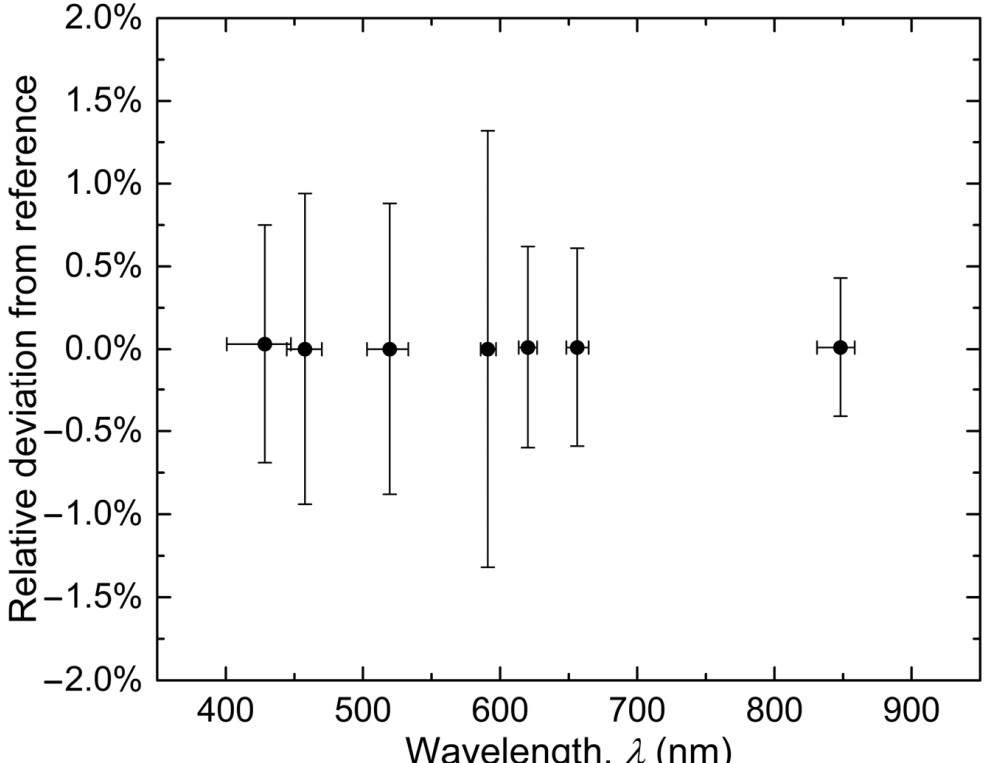

**Figure 5.** Mean and standard deviation of the deviation from reference for each of the seven bands considered. The horizontal error bars represent the width at half maximum of the emission spectrum of each LED. The standard deviation was considered as the inaccuracy value of each band.

**Table 3.** Inaccuracy for each of the bands studied.

| Band | Inaccuracy (% of Measurement) |
|---|---|
| Blue-I | 0.72% |
| Blue-II | 0.94% |
| Green | 0.88% |
| Orange | 1.32% |
| Red-I | 0.61% |
| Red-II | 0.60% |
| Infrared | 0.42% |

### 3.2. Precision Determination

The precision of our device was determined and compared with that of two commercial devices commonly used to measure the NDVI, namely: the *FieldScout CM 1000 NDVI Meter* (Sprectrum Technologies, Inc., Aurora, IL, USA), whose red and infrared bands are centered at a wavelength of 660 nm and 840 nm, respectively [55], and has a sampling area of approximately 1 cm$^2$, and the *PlantPen NDVI 310 Meter* (Photon Systems Instruments, Drasov, Czech Republic), with red and infrared bands centered at 635 nm and 760 nm [56], and a sampling area of approximately 6 cm$^2$. In both cases, the manufacturers declare a spectral bandwidth of 10 nm for both bands. These devices only give the value of the NDVI and do not provide any additional information about the reflectance spectrum of the plant.

To determine the precision of each instrument, NDVI measurements were performed on 150 plant samples with leaves of different morphology, texture, and color. In the case of our device, the measurements were performed in two ways: without using any cover lid, which provides a sampling spot with a radius of around 16 mm, and using a lid with a 4 mm radius circular opening. The precision, denoting the reproducibility of the measurements, was calculated as the standard deviation of ten successive measurements on each sample, removing and placing the sample again on the instrument every time, normalized to the mean value.

Figure 6 shows a histogram of the standard deviation of each series of ten NDVI measurements performed on each of the samples with each device. The value of the central point of a Gaussian fit to each histogram was taken as the average precision of that device.

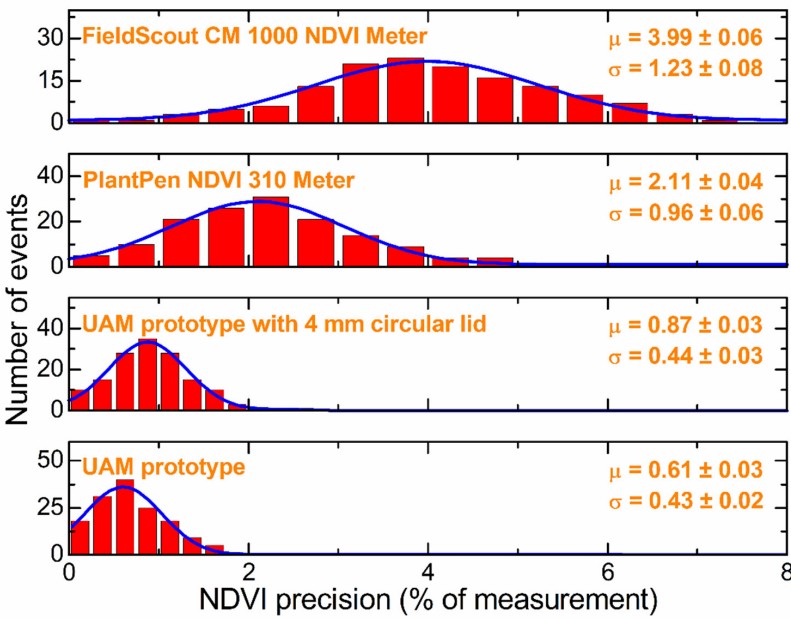

**Figure 6.** Histograms of the standard deviation between measurements performed with our prototype, with a 4 mm circular lid and without any lid, and with two commercial devices. The blue line indicates the Gaussian fit to the data. The precision was taken as the central point of the Gaussian fit.

The NDVI values obtained with our prototype present an average dispersion between measurements of 0.61% when no lid was used, and 0.87% when a lid with a 4 mm radius circular opening was used. These values are noticeably better than those of the two commercial devices, which showed a dispersion higher than 2% for the *PlantPen NDVI 310 Meter*, and almost 4% for the *FieldScout CM 1000 NDVI Meter*.

The reason why our prototype shows a higher precision without a lid than with it can be attributed to that, when measuring without a lid, a larger area is sampled, providing a more evenly averaged measurement. Accordingly, when the 4 mm radius lid was used, the measurement uncertainty increased by about 40%. Even in this worst case, the precision of our device is still better than that of the commercial instruments, despite those having a larger sampling area. This suggests that the signal processing performed in our case has a noticeable influence on the overall performance of the device.

### 3.3. Signal-to-Noise Ratio Improvement

To further investigate the above, the accuracy and precision of our prototype were studied before and after each filtering stage (LPF, adaptive amplification, and DSF), to assess how each one affects the overall performance of the instrument. The goal was to minimize the noise floor (NF) of the instrument in order to maximize the signal-to-noise ratio. Table 4 shows the average inaccuracy, the NDVI precision, and the NF for each amplification circuit after the implementation of each of the signal processing stages with no lid attached. With no signal processing (the bare signal from the photodiode) the discrepancy between the measurement and the reference is almost 70%, with a precision in NDVI determination of 44% and a NF of 6% full scale. These values improve with the LPF stage, obtaining 26%, 32%, and less than 4%, respectively, and most noticeably after the adaptive amplification, when the inaccuracy drops to 5.4% and the precision to less than 10%, while the NF decreases to less than 1.5%. The numerical DSF further improves the performance of the device, reaching a final average inaccuracy of 0.78%, a precision of 0.61%, and a NF of less than 0.10%. It is noteworthy that with such a simple design, the inaccuracy of this prototype is barely twice the inaccuracy of high-end laboratory spectrophotometers, which typically have accuracies of around 0.3% full scale.

**Table 4.** Average inaccuracy, NDVI precision, and NF for each of the amplification circuits after the implementation of each processing stage.

| Stages | Inaccuracy (% of Measurement) | NDVI Precision (% of Measurement) | NF (% of Full Scale) | | |
|---|---|---|---|---|---|
| | | | Gain 1 | Gain 2 | Gain 3 |
| No Signal Processing | 68% | 44% | 6.0% | -- | -- |
| LPF | 26% | 32% | 3.4% | -- | -- |
| LPF + Amplification | 5.4% | 7.7% | 0.74% | 1.0% | 1.2% |
| LPF + Amplification + DSF | 0.78% | 0.61% | 0.06% | 0.08% | 0.10% |

Note that the NF of the amplifier circuit is higher the higher the gain. This is consistent with what is expected, since, as the amplification is higher, any unwanted noise components are also being amplified.

### 3.4. Comparison with Commercial Devices

An analysis of the consistency between the results obtained using our device and the results obtained using commercial instruments was carried out. For this, the average NDVI values obtained using our device and using the two commercial devices were noted for each sample, as well as their respective errors.

Figure 7 shows the NDVI values obtained using the commercial devices in relation to the value obtained with our device. The results obtained using our prototype and each of the commercial devices are consistent, which is observed from the linear relationship (linear correlation factor greater than 0.999 in both cases) between the NDVI index measured with

commercial devices and with our prototype. In both cases, the slope is almost unity: 1.008 for the *FieldScout CM 1000 NDVI Meter*, and 1.044 for the *PlantPen NDVI 310 Meter*.

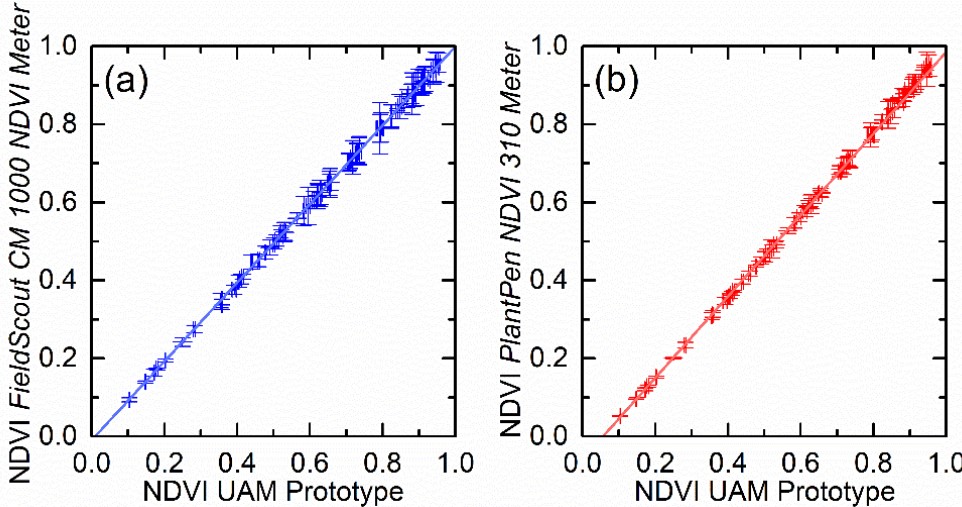

**Figure 7.** NDVI value obtained with the commercial instruments (**a**) *FieldScout CM 1000 NDVI Meter* and (**b**) *PlantPen NDVI 310 Meter*, in relation to the NDVI value obtained with our prototype. A linear fit was performed. Error bars represent the standard deviation between measurements.

The fact that the *PlantPen NDVI 310 Meter* device gives results that differ more from ours is because the bands it uses to calculate the NDVI are somewhat more different than ours than those of the *FieldScout CM 1000 NDVI Meter* instrument, which practically coincide with ours [55,56]. This indicates that when using these instruments, it is important to always use the same model, or have calibrations performed for different models, since measurements can vary slightly between devices.

In both cases, it can be seen how the precision of the measurements, represented by the error bars, is better for our device, which is consistent with what was said above.

## 4. Experimental Tests

To analyze the possible applications of the implemented prototype, as well as to compare the features it offers in relation to commercial devices, some potential use cases were studied by two experimental trials, namely: the sensitivity of the device to changes in the sunlight exposition and soil water conditions of a plant, as well as to the level of fruit ripening, were studied.

### 4.1. Sensitivity of the Device to Changes in Sunlight Exposure and Soil Water

The foliar variation in the NDVI with the level of soil water and solar exposure was studied for several individuals of *Alocasia odora*. This species was chosen because it has large and leathery leaves, which allows several measurements to be made without damaging the leaf surface, as well as because its large and monochromatic leaves allow taking measurements without introducing errors due to possible variations in the color of the plant. Nine adult individuals of similar size and age were exposed to the same level of incident sunlight for two months (5 June–4 August), in a semi-shade location where no direct sunlight fell on the plants at any time. During this time, the soil water level was the same for all plants, remaining at field capacity.

The following experimental design was carried out: a sample of three plants was moved to a position where the sunlight fell directly (maximum solar radiation), another sample (*n* = 3) was moved to a completely dark room (no solar radiation), and the remaining sample was left in its original location (semi-shade). The plants were kept in these locations during the rest of the experiment. Throughout the experiment, the plants were watered daily at the same time of the day, so the soil water level was maintained at field capacity.

During the following twenty days, measurements of the reflectance spectrum were taken daily always at the same time (23:30 local time). For this, five successive measurements were performed on three leaves of each of the plants in each sample. Subsequently, the average NDVI value for the plants subjected to each of the sunlight conditions studied was calculated.

On the other hand, to analyze the dependence of the NDVI on the soil water level, the same experimental design was carried out, modifying in this case the soil water level, as follows: another nine plants were taken, and the same sample preparation process described above was carried out for two months. After this time, a sample (three plants) continued to be irrigated at field capacity, another sample was irrigated with triple the amount of water (high capacity), and another sample was not irrigated during the rest of the experiment (low capacity). During the experiment, the plants remained in the same semi-shade position, so that the conditions of exposure to sunlight were not modified. Measurements were performed daily, for twenty days, using the same procedure described in the previous experiment.

These studies were carried out both with our prototype and with the *PlantPen NDVI 310 Meter* commercial device, in order to analyze the features offered by each device.

Figure 8a shows the evolution of the NDVI for the *Alocasia odora* plants as a function of the level of soil water, while Figure 8b shows the evolution of the NDVI as a function of the level of exposure to sunlight.

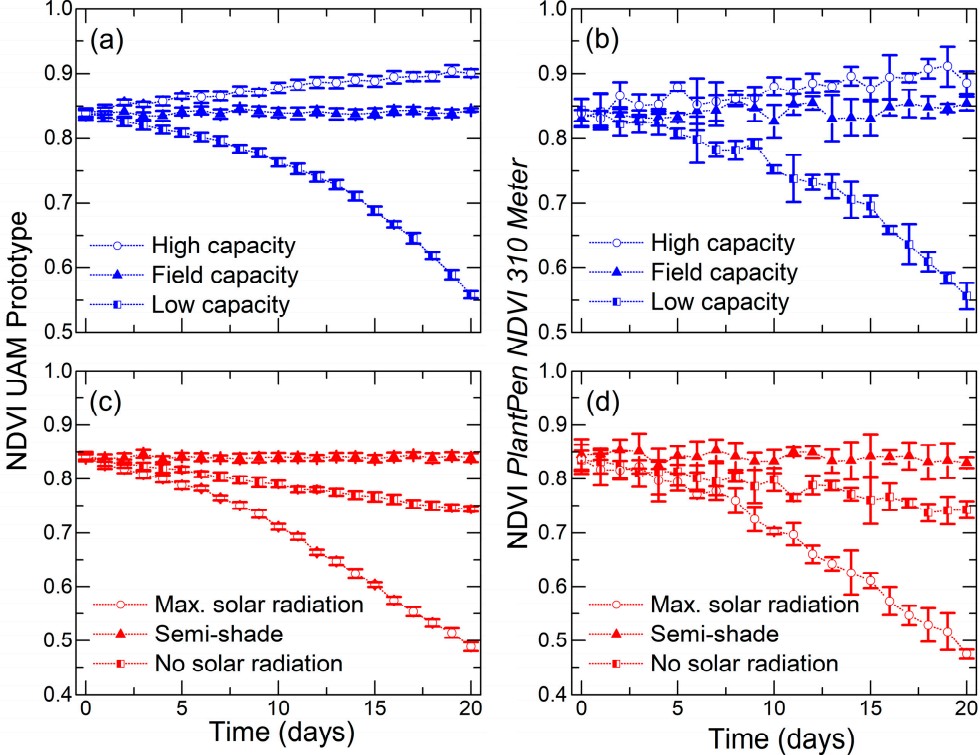

**Figure 8.** Evolution of the foliar NDVI index of *Alocasia odora* plants when the conditions of (**a**,**b**) watering and (**c**,**d**) exposure to sunlight were modified. The results obtained with (**a**,**c**) our prototype and with (**b**,**d**) the commercial device *PlantPen NDVI 310 Meter* are shown. The vertical error bars represent the deviation between the various measurements made.

It can be observed that if the level of soil water increases, the NDVI value increases, although the variation is small. This indicates that there is saturation at the tissue level and that the plant does not absorb more water, even if there is water in the soil. In the entire experiment, no perceptible changes to the human eye could be observed in the plants when the watering level was increased. On the other hand, it was observed that when there is no irrigation, the NDVI index decreases considerably, which indicates that the physiological

state of the plant is worsening. In this case, a perceptible change to the human eye could be observed in the plants after the fifth day of the experiment, which indicates a very negative foliar osmotic pressure and, consequently, produces permanent wilting within a few days.

Regarding the evolution of the NDVI with the level of exposure to sunlight, it can be observed that whether the level of solar exposure increases or if the plant is not allowed to receive any sunlight, the value of the NDVI decreases. This indicates that the plant's photosynthetic efficiency is decreasing, and therefore its physiological state is deteriorating. It is consistent with what is expected that the physiological state of the plant deteriorates if the level of exposure to sunlight is very high, since the *Alocasia odora* plant is a shade plant, which does not tolerate being constantly exposed directly to sunlight [57–59]. On the other hand, this plant can survive quite a long time in a very low-light environment [58], which is why the NDVI value of the plant when placed in a dark room did not decrease as drastically as when placed in a full sun area.

To be able to observe an appreciable variation in the NDVI, it is necessary that the variation produced is greater than the uncertainty of the measurement made. In cases where the NDVI index decreases, and therefore the physiological state of the plant is worsening, no visible changes were observed in the plants until four to five days after the change in conditions. However, our device allows us to know that the value of the NDVI index is decreasing from the following day or the two days following the change in plant conditions, which allows us to take the necessary actions so that the plant's physiological state stops deteriorating faster.

Likewise, it is observed that the evolution of the NDVI with the change of sunlight exposure or soil water conditions is smoother and more uniform when the measurement is performed with our device. All this is due to the low level of uncertainty of our device with respect to the commercial one.

Due to the above, it would be possible to appreciate in agricultural exploitations, garden centers, etc., that a change in the physiological state of the plants is taking place more quickly by using our prototype. This would allow us to act accordingly more quickly before changes in the plant are perceptible to the human eye.

### 4.2. Sensitivity of the Device to Fruit Ripening

The dependence of the reflectance spectrum of round tomatoes (*Solanum lycopersicum*), strawberries (*Fragaria vesca*), and conference pears (*Pyrus communis*) on their level of ripening was analyzed. To carry this out, five fruits of each type that were as unripe as possible, with a reflectance spectrum that initially was very similar for all of them (variations of less than 1.5% for each one of the analyzed bands) were taken.

Measurements of the reflectance spectrum of the different fruits were carried out for fifteen days. The fruit was preserved in a refrigerator at 5 °C during the duration of the experiment. For each piece of fruit, three different points were considered, but in such a way that measurements were performed on the same points every day. In this way, errors due to possible inhomogeneities in the color of the piece of fruit were avoided. Five measurements were performed on each point. The reflectance spectrum of a certain type of fruit after certain days of ripening was calculated from the average of all the measurements performed.

This study was carried out both with our prototype and with the *FieldScout CM 1000 NDVI Meter* commercial device, to analyze the features offered by each device. It is worth mentioning that was is not possible to carry out this study with the *PlantPen NDVI 310 Meter* commercial device because, due to its geometry, measurements can only be made on flat samples.

Figure 9a,c,e shows the reflectance spectra obtained for tomato, strawberry, and pear on the day the experiment started and five, ten, and fifteen days later. In all cases, a variation in the reflectance spectrum of the sample is observed. In the case of tomatoes, there is a variation in the reflectance of all the studied bands, with this variation being more pronounced in the blue and green regions. In the case of strawberries, there is a very

notable variation in the blue and green bands, while in the red and infrared bands, the reflectance hardly varies. Finally, in the case of pears, the infrared region is the only one in which an appreciable variation in the reflectance spectrum is observed.

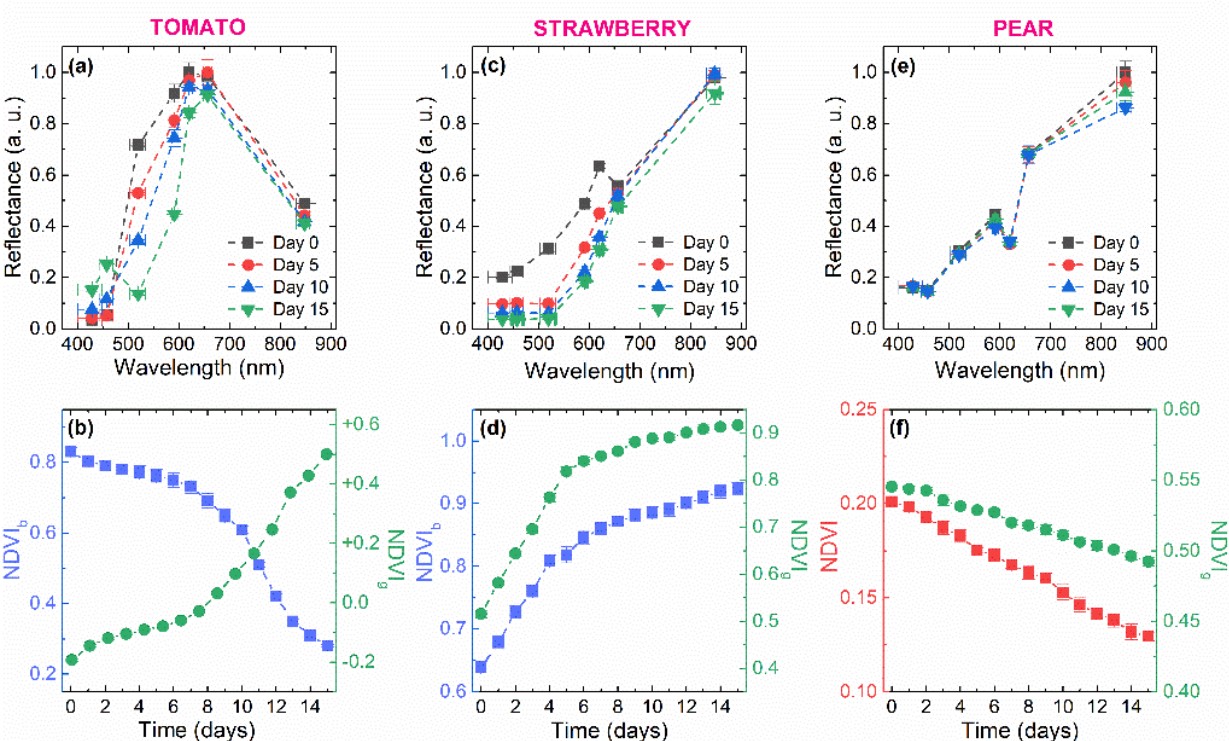

**Figure 9.** Reflectance spectrum of (**a**) round tomato, (**c**) strawberry, and (**e**) conference pear on the day the experiment was started, and five, ten, and fifteen days later. Evolution of different vegetation indices for (**b**) round tomato, (**d**) strawberry, and (**f**) conference pear as a function of the days elapsed since the start of the experiment.

When using an index to characterize the dependence of the reflectance spectrum on the fruit ripening level, it is convenient to use a band in which the reflectance does not vary much during the ripening process and a second band where it does vary markedly and progressively during the process, so that changes in the index are also progressive during ripening.

Thus, because in the case of tomatoes and strawberries, the variation of reflectance in the red and infrared band is small in both cases, it is not pertinent to use the NDVI to characterize the degree of ripeness of the fruit. However, since for both fruits the reflectance in the blue and green bands does vary markedly with ripening, it is possible to use the $NDVI_b$ and $NDVI_g$ indices to characterize the degree of ripening of the fruit. In the case of tomatoes, since the signal in the green band is notably larger and undergoes greater variations, of the two indices, the $NDVI_g$ index shows a greater variation. In the case of strawberries, the variation of both indices with ripening is similar. This is shown in Figure 9b,d.

On the other hand, in the case of pears, since the reflectance only varies appreciably in the infrared band, this band must be necessarily considered, in addition to any other band. Therefore, the NDVI, $NDVI_b$, and $NDVI_g$ could be applicable. Figure 9f shows the evolution of the NDVI and $NDVI_g$ indices as a function of the days elapsed since the start of the experiment, although the $NDVI_b$ index could have also been considered.

It is appropriate to remember that our instrument, unlike commercial devices, can acquire the reflectance value in seven bands, as well as compute a greater variety of indices, which allows us to consider in each case the most pertinent index for one particular application. In the case of commercial instruments, only one vegetation index is measured:

the NDVI. Therefore, it was not possible to analyze in any way the degree of tomato or strawberry ripening with the two commercial instruments considered.

Likewise, the fact that our device, thanks to the different lids, can be used with objects with different geometries and sizes, allows the performance of this type of study, which was impossible by using the *PlantPen NDVI 310 Meter* commercial device.

Figure 10 shows the evolution of the NDVI of the conference pear with the number of days elapsed since the beginning of the experiment, measured both with our prototype and with the *FieldScout CM 1000 NDVI Meter* commercial device.

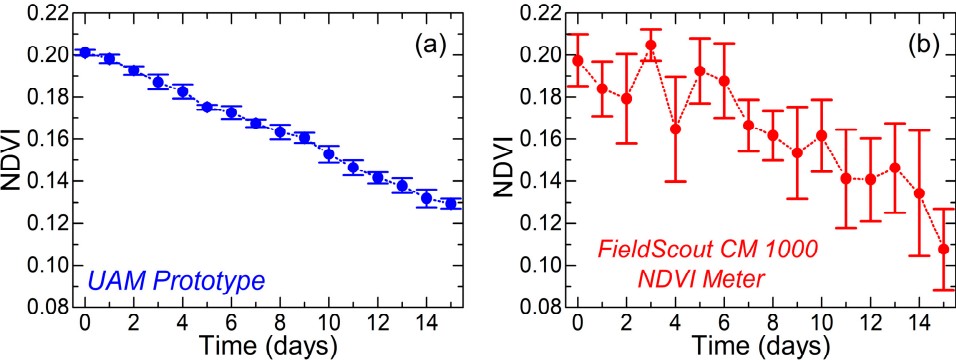

**Figure 10.** Evolution of the NDVI of conference pear as a function of the number of days elapsed since the beginning of the experiment. The results obtained with (**a**) our prototype and with (**b**) the commercial device *FieldScout CM 1000 NDVI Meter* are shown. The vertical error bars represent the deviation between the various measurements made.

To determine that a change is taking place in a certain index because of the ripening of the fruit, the variation in the index must be greater than its uncertainty value. In the case of our device, it is possible to monitor the ripening of fruits from one day to the next, even before changes are perceptible to the eye. On the other hand, when using the commercial instrument, this margin of error is much greater, of the order of a week, when the changes can already be seen without the need for spectrophotometric measurements. Again, the best performance of our device is due to its lower level of uncertainty.

This approach could have important industrial applications, such as being able to determine the optimal moment at which fruits should be picked from the tree in a very economical and simple way.

## 5. Conclusions

In the present work, a spectrophotometer based on the open-source Arduino platform was implemented, covering seven bands from blue to infrared and able to determine up to nine different vegetation indices.

An accuracy between 0.4% and 1.3% was obtained in the measurement of the reflected light intensity for the different bands. The precision or repeatability in the measurement of the NDVI was determined, obtaining a value of 0.61%. The precision value obtained for our prototype is notably better than those of the two commercial instruments used, which present a repeatability of 2.11% and 3.99%. It was found that the implementation of the LPF stage, the adaptive amplification stage, and the DSF were necessary to obtain such good accuracy and precision values. Therefore, the better performance of our device with respect to commercials may be because commercial devices may not have implemented some of these processing stages.

The applicability of the device to study changes in the physiological state of plants when their sunlight exposure or soil water conditions are modified, as well as to monitor the ripening state of different fruits has been proven. In both cases, the results obtained with our prototype have been more satisfactory than those obtained with commercial devices, and have made it possible to observe changes in the reflectance spectrum before the changes in the plant or the fruit were perceptible to the bare eye.

The high accuracy and precision of this device, as well as its low cost and versatility when it comes to being used on objects other than leaves, such as fruits or vegetables, justify the scientific relevance of this work. In addition, the way to assemble and program the device was explained in detail, so that it can be reproduced adapting to the needs of each experiment. In this way, access to this low-cost technology that can be used to optimize the production of agricultural holdings is favored for farmers in developing countries, contributing to the fulfillment of Sustainable Development Goals 2 (Zero Hunger) and 12 (Responsible Consumption and Production)

**Author Contributions:** F.J.F.-A. designed and built the instrument described in the article and performed all measurements to determine its precision and accuracy. F.J.F.-A. designed and carried out the experiments with plants and fruits and collected and analyzed the results. F.J.F.-A. also wrote the first and main draft of the manuscript, gained access to commercial comparison tools, and created and formatted all graphs. Z.H. discussed the usefulness of this type of instrument, helped to interpret some of the results of the experimental tests, and made comments and corrections to the draft of the manuscript. V.T.-C. raised the initial design of the instrument, and supervised the work, providing the necessary means and equipment for its development. V.T.-C. also made comments and corrections to the manuscript. All authors have read and agreed to the published version of the manuscript.

**Funding:** This work was funded by the Comunidad de Madrid, Spain, within the framework of the agreement with the Universidad Autónoma de Madrid, Spain in item "Excellence of University Professorate".

**Institutional Review Board Statement:** Not applicable.

**Data Availability Statement:** Data are available from the corresponding author upon reasonable request.

**Acknowledgments:** The authors wish to thank Luis G. Pelayo for his technical assistance, and J.M.G. for lending the commercial devices used for this study. Moreover, this work has been awarded in the "II Call for Awards for Research in the Field of the 2030 Agenda of the Universidad Autónoma de Madrid" as the best bachelor's thesis.

**Conflicts of Interest:** The authors have no conflict of interest to declare.

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
