# Peer review of "A Cost-Effective Portable Multiband Spectrophotometer for Precision Agriculture"

_agriculture, doi:10.3390/agriculture13081467_

Round 1

Reviewer 1 Report

This paper proposes a promising low-cost, portable multi-frequency spectrometer for precision agriculture applications. However, there are several issues that need to be addressed in the subsequent version of this paper, including expanding wavelength coverage, enhancing sensing element stability, conducting field validation studies, documenting data processing methods, and addressing calibration drift issues. Addressing these issues would enhance the reliability and generalizability of the spectrometer for practical agricultural applications.

1. The use of only seven wavelength bands in the spectrometer is a limitation, as it may not provide a complete spectral profile of the sample. Future revisions could investigate or discuss the addition of more wavelength bands to improve the accuracy of the instrument.

2. The use of photo diodes as the primary sensing element may introduce errors due to their sensitivity to temperature and light intensity fluctuations. Future revisions could consider or discuss using more stable sensing technologies, such as fiber optic sensors, to improve accuracy and stability.

3. The lack of in-field validation studies limits the generalizability of the results to real-world agricultural settings. Future revisions should include or discuss field trials to validate the performance of the spectrometer in different environmental conditions.

4. The device was tested on a limited number of sample types and sizes, raising concerns about its versatility across different agricultural substrates. Future work should include testing on a wider range of sample types and sizes to confirm the device's performance.

5. The paper lacks a detailed description of the data processing algorithms used, making it difficult to reproduce the results. Future revisions should include a comprehensive description of the data processing methods to facilitate replication studies.

6. The use of reflectance values as a measure of sample properties may not be appropriate for certain applications, such as monitoring plant stress response or disease progression. Future revisions could investigate alternative metrics or spectral indices that better capture these processes.

7. The device was designed with portability and low cost in mind, but the overall size and weight of the spectrometer may limit its practicality for field applications. Future revisions could explore the use of adequate components or more compact designs to improve usability in remote or rural areas by UAV (Unmanned Aerial Vehicle) or UGV (Unmanned Ground Vehicle).

8. The paper does not address the issue of calibration drift over time, which can affect the accuracy of the instrument over extended periods of use. Future revisions should consider developing drift compensation strategies or regular calibration procedures to ensure long-term stability of the system.

This paper is generally well-written, and the authors present their ideas in a coherent and logical manner. The authors have a clear understanding of the topic and provide a detailed background on the subject, setting a solid foundation for their research.

However, there are a few areas where the grammar and sentence structure could be improved to enhance the clarity and readability of the paper. Some sentences are a bit long and complex, which can make the paper more difficult to read and understand. Breaking these sentences down into simpler, more concise sentences could help to clarify the authors' points and make the paper more accessible to readers.

There are also a few instances where the authors use informal or less precise language, such as "building blocks" and "the naked eye." While these phrases are understandable, they might not convey the authors' points as accurately or professionally as more formal language would. Replacing these phrases with more precise, scientific language could improve the overall tone and professionalism of the paper.

Author Response

Dear reviewer,

Thank you for the time you have taken to review our manuscript, as well as for the many constructive comments you have made on our work. In the attached file you will find the response to each of the indicated points.

Reviewer 2 Report

This article presents the development of a cost-effective portable multiband spectrophotometer for precision agriculture. The instrument is equipped with a microprocessor that allows for the acquisition of reflectance spectra and the computation of vegetation indices. The accuracy of reflectance measurements is comparable to high-end spectrophotometers, and the precision in determining the normalized difference vegetation index (NDVI) is significantly better than more expensive commercial instruments.

According to my expertise, this research is excellent and well-reported in this manuscript. The introduction contains research problems in a very detailed and comprehensive manner. The methodology is presented in great detail for step-by-step procedures. The research results have been discussed and compared with previous studies. Finally, the conclusions have answered the hypotheses and objectives of this study.

Therefore, without hesitation, I express that this article can be acceptable. However, I have small suggestions that the Authors can consider to complement this article better, including:

# Please to check the format of references guideline in this manuscript with the rules of a journal.

# Please to consider to be able to arrange the chapters of this manuscript to be more general. Generally, research articles contain Introduction, Materials and Methods, Results and Discussion, Conclusions, and References. Please to include a similar chapter in the main chapter.
Thank you, and all the best.

This article presents the development of a cost-effective portable multiband spectrophotometer for precision agriculture. The instrument is equipped with a microprocessor that allows for the acquisition of reflectance spectra and the computation of vegetation indices. The accuracy of reflectance measurements is comparable to high-end spectrophotometers, and the precision in determining the normalized difference vegetation index (NDVI) is significantly better than more expensive commercial instruments.

According to my expertise, this research is excellent and well-reported in this manuscript. The introduction contains research problems in a very detailed and comprehensive manner. The methodology is presented in great detail for step-by-step procedures. The research results have been discussed and compared with previous studies. Finally, the conclusions have answered the hypotheses and objectives of this study.

Therefore, without hesitation, I express that this article can be acceptable. However, I have small suggestions that the Authors can consider to complement this article better, including:

# Please to check the format of references guideline in this manuscript with the rules of a journal.
# Please to consider to be able to arrange the chapters of this manuscript to be more general. Generally, research articles contain Introduction, Materials and Methods, Results and Discussion, Conclusions, and References. Please to include a similar chapter in the main chapter.

Thank you, and all the best.

Author Response

Dear reviewer,

Thank you very much for the time spent reviewing our manuscript, and for the positive comments made on our manuscript. Below you will find the response to each of the indicated points.
